# Glucose-Responsive Silk Fibroin Microneedles for Transdermal Delivery of Insulin

**DOI:** 10.3390/biomimetics8010050

**Published:** 2023-01-24

**Authors:** Guohongfang Tan, Fujian Jiang, Tianshuo Jia, Zhenzhen Qi, Tieling Xing, Subhas C. Kundu, Shenzhou Lu

**Affiliations:** 1National Engineering Laboratory for Modern Silk, College of Textile and Clothing Engineering, Soochow University, Suzhou 215123, China; 23Bs Research Group, I3Bs Research Institute on Biomaterials, Biodegrabilities, and Biomimetics, Headquarters of the European Institute of Excellence on Tissue Engineering and Regenerative Medicine, University of Minho, 4805017 Barco, Portugal

**Keywords:** silk, microneedles, glucose responsiveness, insulin, diabetes

## Abstract

Microneedles (MNs) have attracted great interest as a drug delivery alternative to subcutaneous injections for treating diabetes mellitus. We report MNs prepared from polylysine-modified cationized silk fibroin (SF) for responsive transdermal insulin delivery. Scanning electron microscopy analysis of MNs’ appearance and morphology revealed that the MNs were well arranged and formed an array with 0.5 mm pitch, and the length of single MNs is approximately 430 μm. The average breaking force of an MN is above 1.25 N, which guarantees that it can pierce the skin quickly and reach the dermis. Cationized SF MNs are pH-responsive. MNs dissolution rate increases as pH decreases and the rate of insulin release are accelerated. The swelling rate reached 223% at pH = 4, while only 172% at pH = 9. After adding glucose oxidase, cationized SF MNs are glucose-responsive. As the glucose concentration increases, the pH inside the MNs decreases, the MNs’ pore size increases, and the insulin release rate accelerates. In vivo experiments demonstrated that in normal Sprague Dawley (SD) rats, the amount of insulin released within the SF MNs was significantly smaller than that in diabetic rats. Before feeding, the blood glucose (BG) of diabetic rats in the injection group decreased rapidly to 6.9 mmol/L, and the diabetic rats in the patch group gradually reduced to 11.7 mmol/L. After feeding, the BG of diabetic rats in the injection group increased rapidly to 33.1 mmol/L and decreased slowly, while the diabetic rats in the patch group increased first to 21.7 mmol/L and then decreased to 15.3 mmol/L at 6 h. This demonstrated that the insulin inside the microneedle was released as the blood glucose concentration increased. Cationized SF MNs are expected to replace subcutaneous injections of insulin as a new modality for diabetes treatment.

## 1. Introduction

Diabetes is considered one of the world’s most complex health problems in the 21st century and a severe chronic disease [1]. Blood sugar levels rise when the pancreas cannot make insulin or the body cannot fully use its insulin. High blood sugar levels can affect the heart, vessels, eyes, and other vital organs, causing severe illnesses [2]. The therapeutic effectiveness of insulin was first demonstrated in 1921, and insulin in subcutaneous injections has been the basis of Insulin therapy [3,4]. This method is often uncomfortable for the patient. Frequent injections can also increase pain and wounds caused by needles also increase the risk of infection [5,6]. Realizing the use of insulin through non-injection routes has become a new direction of human research [7].

The transdermal drug delivery system [8], one of the driving areas of drug delivery research, offers significant advantages over oral and injectable drug delivery [9,10]. As a third-generation drug delivery system [11], MNs can deliver drugs to the dermal area of the skin [12,13,14], which provides a new idea for transdermal drug delivery [5].

The original insulin delivery MNs came from drug coatings [15]. As a delivery vehicle, MNs create microchannels in the skin [12]. The drug contained in the layer can enter the body seamlessly, though the coating thickness and needle size limit the drug-carrying capacity. Later, dissolving MNs were developed, which improved the utilization of MNs loading. Microneedles made of water-soluble, biocompatible polymers dissolve or degrade upon successful puncture into the skin, releasing the drug from internal storage instantaneously [16,17]. Chuang et al. [18] prepared a layered MNs system to understand the dissolution pattern. The drug is sealed in the MNs tip. By what means does the MNs carrier leave the body after dissolution, whether it will impact the human body and so on is still unknown. Therefore, ensuring how the MNs substrate dissolves becomes a new goal [19]. 

Hydrogel-forming microneedles is an exciting and recent development in microneedling technology. Donnelly et al. [20] first identified that hydrogel could be used as a MNs base material. Upon insertion into the skin, the hydrogel microneedles swell in the presence of tissue fluid due to the inherent hydrophilicity of the polymer, enabling it to act as a delivery device for drugs into the skin. The gel form ensures the continuity of the medicine from the reservoir to the dermal microcirculation, making long-term transdermal drug delivery possible [21]. Behl et al. [22] prepared a memory-freezing gel with glucose-responsive gel volume transformation using the reversibility of borate lipid bonds. Chen et al. [23] reported a semi-interpenetrating network hydrogel. By preparing a double layer of MNs, it is ensured to keep the original needle shape during the drug release process.

SF is a typical biological material with various applications, including drug delivery and scaffold synthesis in multiple forms, particularly in hydrogel [24,25,26]. In addition, SF has excellent characteristics, better biodegradability, low inflammatory response, low immunogenicity, and high mechanical properties [27,28]. In the previous studies, SF was introduced as a base material for MN systems to encapsulate or support the application of medication [29,30] because it has the following advantages: (1) Maintains protein stability after long processing times; (2) Ability to eliminate the effects of shear stress and temperature stress on encapsulated drugs. Meanwhile, in solution conditions, SF can be easily and quickly processed into MNs without using organic solvents, ultraviolet cross-linking, and other auxiliary means [30]. The β-sheet crystal content in the secondary structure plays an essential role in achieving mechanical strength and characteristic deformation for tuning mechanical properties and release kinetics. Tuning the β-sheet range using methanol vapor annealing can aid in rapid and sustained drug release [28,31,32]. Rojas et al. [33] blended PVA with SF to prepare polymer MNs, which can improve the transdermal delivery of porphyrins. Yin et al. [34] developed an SF MNs drug delivery system with different small molecule reagents, with a tremendous transdermal drug release effect and little dissolution loss. Qi et al. [35] developed composite MNs based on proline-modified SF. In vitro studies have shown that MNs can achieve sustained drug release. Therefore, using modified SF as a substrate to achieve MNs solubilization but not dissolution is a reliable solution.

Due to the slow degradation and swelling of the polymer, MNs need to release the encapsulated drug over a more extended period [36]. According to the patient’s application environment, it is necessary to remove sealed drugs quickly, selectively, and deliver controlled medications to achieve high therapeutic efficiency [16]. Xu et al. [37] reported an intelligent pH-responsive transdermal delivery system that controls the release of internally sealed insulin by covering bioactive glass nanopores with ZnO quantum dots. Jiang et al. [38] prepared mesoporous bioactive glass nanoparticle MNs modified by a multifunctional enzyme layer, which adjusted the local pH through Gox and disrupted the compound enzyme layer to release preloaded insulin. However, relatively few studies have been conducted on intelligent responsive SF MNs for controlled drug release.

In order to meet the daily insulin dose requirement of diabetic patients and avoid the risk of hypoglycemia caused by insulin overdose, an insulin-loaded modified SF MNs with both high dose and intelligent drug delivery effect was developed in this paper. In the previous stage, our research group used polylysine to modify SF and prepared SF hydrogel sensitive to acidic pH values, the swelling degree of which is different under different pH values [39]. Based on the previous study’s results, glucose-responsive MNs were designed by introducing glucose oxidase (Gox) in this paper. The MN-encapsulated enzyme converts the glucose concentration signal into a pH signal. This stimulates the MNs to develop a lysis response to glucose, enabling MNs to self-regulate insulin delivery.

## 2. Materials and Methods

### 2.1. Experimental Materials

Silkworm cocoons were purchased from Suzhou Xiancan silk Biotechnology Co., Ltd. (Suzhou, China). Horseradish peroxidase (HRP), Gox, and human recombinant insulin (100 IU/mg) were obtained from Shanghai Alading Biotechnology Co., Ltd. (Shanghai, China). An insulin detection kit (chemiluminescence immunoassay (CLIA)) was acquired from Henan Meikai Biotechnology Co., Ltd. (Henan, China). Male SD rats were provided by the Experimental Animal Center of Suzhou University, (Suzhou, China). The Ethics Committee of Soochow University approved all animal experiments, which followed the Guidelines for Ethical Conduct in the Care and Use of Research Animals established by Soochow University. All other chemicals used in the experiments were analytically pure. Ultra-pure deionized water was used for all experiments.

### 2.2. Preparation of SF Solution

As shown in Figure 1a, cocoons were put into 4000 mL, 0.1 wt% sodium bicarbonate (NaHCO_3_), 0.3 wt% sodium carbonate (Na_2_CO_3_) solution, heated to boiling, removed after 30 min, and cleared of silk glue using deionized water. The above experimental steps were repeated three times. After drying SF in an oven at 60 °C, SF was dissolved in 9.3 mol/L lithium bromide solution for about 1 h, cooled, and put into dialysis bags, dialyzed with deionized water for 3–4 d, then removed and centrifuged to obtain the filamentous solution.

### 2.3. Preparation of Cationized SF [39]

As described in Figure 1b, SF solution with a concentration of 30 mg/mL was prepared in a 50 mL beaker. The beaker was stabilized in an ice bath environment, and 0.1 mol/L of million-methanesulfonate (MES) solution was added in small amounts several times to adjust the SF solution to about 5.5. Slowly add 0%, 1%, 2%, 5%, 10% and 15% polylysine solution of SF. Add Sodium bicarbonate N-hydroxysuccinimide (NHS) (10 mg/mL) with 2.5% of SF quantity. Add 1-Ethyl-3 (3-dimethylamino propyl) carbodiimide (EDC) with 5% of SF weight. React in the ice bath for 4 h, remove, and keep in the refrigerator overnight at 4 °C. After that, the solution was dialyzed with deionized water for 3 days. The supernatant solution was centrifuged to obtain the grafted SF solution. In the MNs used in animal experiments, 10% polylysine was added to the cationic-modified SF substrate.

### 2.4. Fluorescent Labeling of Human Insulin

Figure 1c shows that human insulin was dissolved in a pH = 9.0 carbonate buffer with a concentration of 0.1 M (containing 0.2 mmol/L ethylene diamine tetraacetic acid to prevent insulin aggregation), resulting in a solution concentration of 20 mg/mL. Isothiocyanic acid fluorescein isomer I (FITC) was dissolved in dimethyl sulfoxide at 5 mg/mL concentration. The two solutions were mixed in a magnetic agitator and stirred in a light-free environment for 12 h. Then, they were put into a 3 kDa dialysis bag, and dialysis was carried out for 2 d successively in carbonate buffer solution with pH = 9.0 and a concentration of 0.1 M and deionized water. Finally, the solution was freeze-dried to obtain fluorescent-labeled human insulin (FITC-INs).

### 2.5. Preparation of Glucose-Responsive SF MNs

In the cationized SF solution (polylysine-modified solution accounting for 10% of the SF quantity), add HRP with a concentration of 10 IU/mL, Hydrogen peroxide (H_2_O_2_) with a concentration of 1 mM, and Gox with a concentration of 0, 2, 4, and 6 mg/mL, respectively. FITC-INs were added to the mixed solution to ensure the insulin content was 10 IU/tablet.

The hybrid solution was poured into a polydimethylsiloxane (PDMS) mold and pumped into a vacuum-drying oven [34]. The mixed solution was formed into a hydrogel in a sealed environment and then dried in a constant temperature and humidity environment. MNs were obtained after removal from the mold. The production process is shown in Figure 1d. Among the MNs used for animal experiments, the content of Gox was 4 mg/mL, and the range of Ins was 10 IU/tablet. Detection was performed with a CLIA kit.

### 2.6. Characterization of Cationized SF

#### 2.6.1. Zeta Potential

Pipette 1 mL each of cationized and pure SF solution (concentration 10 mg/mL) to the measurement cell of Malvern ZetasiZer Nano ZS90 particle size potential meter (Malvern, Malvern Hills, United Kingdom). Each sample was measured three times at a controlled temperature of 25 °C to take the average value as the final result.

#### 2.6.2. Fourier Transform Infrared Absorption Spectroscopy (FTIR)

The cationized SF solution was frozen in liquid nitrogen and dried in a freeze-dryer for 48 h. After that, the sample was processed and cut into powder form with scissors, and then the required sample was prepared by the KBr compression method. The absorbance in the range of 400–4000 cm^−1^ was scanned on a Nicolet 5700 FT-IR Fourier transform infrared spectrometer (Thermo Elemental, Waltham, Commonwealth of Massachusetts, United States) as required to obtain the infrared absorption spectra.

### 2.7. Morphology of SF MNs

An MZFL III fluorescence microscope (Leica, Weztlar, Germany) was used in observation for the preparation of microinjection. The MNs were put into liquid nitrogen for quick freezing and vacuum drying. Part of the MNs was cut and taken out after spraying gold on the surface for 90 s. The morphology was observed on an S-4800 scanning electron microscope (Hitachi, Tokyo, Japan).

### 2.8. Structural Measurement of SF MNs

#### 2.8.1. FTIR

Glucose-responsive SF MNs and pure SF MNs were assayed according to method Section 2.6.2.

#### 2.8.2. Crystal Structure of SF MNs

Glucose-responsive SF MNs and pure SF MNs were freeze-dried and ground, processed into powder, pressed into a sample holder, and scanned for diffraction intensity profiles between 2θ = 5° and 45° using an X’PERT PRO MPD fully automated X-ray diffractometer (Bruker, Karlsruhe, Germany) with CuKα radiation, a stable and fluctuation-free tube current of 35 mA, a stable and fluctuation-free tube voltage of 40 kV, and a scanning speed of 8°/min as the environment for their determination.

### 2.9. Mechanical Properties

The mechanical testing and insertion analysis techniques used to characterize microneedles differed between groups [40]. The glucose-responsive and pure SF MNs were cut into small pieces, containing three needles with a surgical blade and tested for compressive mechanical properties in a TMS-PRO mass spectrometer (Food Technology Corporation, Sterling, Commonwealth of Virginia, United States). The initial initiation force is 0.03 N, compression speed is 10 mm/min, and compression deformation rate is 80%. The final results were obtained by averaging 12 parallel samples per group.

### 2.10. Effect of pH on MNs Swelling

After blending with serine polylysine, the hydrogel was used as the control group. The prepared hydrogel was placed in a constant temperature and humidity environment, dried into a film for 1 d, weighed, and recorded as M0. The prepared citric acid/disodium hydrogen phosphate buffer solutions of different pH values were added according to the bath ratio of 1:100 after being placed in a constant temperature water bath at 37 °C for approximately 1 day to reach the dissolution equilibrium, removed, and blotted with filter paper. The surface water of the dry gel film was blotted with filter paper, and the wet weight M1 was recorded. The swelling rate was calculated using the swelling rate formula, as follows.
SR = [(M_1_ − M_0_)/M_0_] × 100%(1)

### 2.11. Effect of Glucose Concentration on pH

Since the internal pH of SF MNs is challenging to monitor, the pH of the SF MNs substrate, i.e., the hydrogel, was tested. An appropriate amount of cationized SF solution was taken, HRP and H_2_O_2_ were added to make a total system of 4 mL, and Gox (4 mg/mL) was added to make a hydrogel. The gel was wrapped on the surface of the pH meter probe and placed in a constant temperature with humidity environment for 1 d to dry, and then a dry gel film was made on the pH meter probe. The dry gel on the head of the SF MNs was simulated to test the dynamic changes of pH inside the dry gel. The dry gels were incubated in different concentrations of glucose solutions (0, 100, 200, 400 mg/dL). The pH inside the dry gels was monitored at specified times using a BP 3000 pH meter (Professional Benchtop pH meter, Goettingen, Germany). In contrast, the pH of the external solution was observed at the same time point for comparison. The specified times were: 0, 30, 60, 90, 120, 150, 180, 210, and 240 min.

### 2.12. In Vitro Drug Release Properties

A standard curve was made to measure the insulin content. FITC-INs were dissolved in PBS buffer solution at gradient concentrations of 0.05~1 μg/mL. The fluorescence intensity of the samples was measured at an excitation wavelength of 485 nm. Then, a linear regression curve of the fluorescence intensity measured by fluorescence spectroscopy versus the mass concentration of FITC-Ins was established, and finally, the following equation was obtained.
Y = 6.1120 × 10^7^x + 5861(2)

#### 2.12.1. Insulin Release in Buffered Solutions of Different pH Values

Glucose-responsive SF MNs were pricked into fresh rabbit skin, fixed, and placed in a 12 mL transdermal release cell. The release solution was set as a citric acid/disodium hydrogen phosphate buffer solution with three pH values of 4, 7, and 9. Pure SF MNs without insulin loading was used as a blank control group. Hence, 1 mL of samples was taken at 1, 2, 4, 8, 12, 24 and 36 h for testing, and fresh 1 mL of buffer solution was added after each sample to ensure the total volume of explanation in the release cell was constant at 12 mL.

Fluorescence values of the collected samples (excitation wavelength: 495 nm; emission wavelength: 516 nm; slit width: 3 nm; temperature: 25 °C) were measured using an FM4P-TCSPC fluorescence spectrometer (HORRIBA Jobin Yvon, Edison, NJ, USA), and the data were recorded to calculate the corresponding FITC-Ins concentrations. According to the conversion of 1 IU = 45.5 μg, the unit of drug release concentration was converted from μg to IU and recorded as Ci. The cumulative release rate was calculated and the cumulative release rate curve of MNs with time was depicted according to the following equation. The calculation formula is as follows.
(3)$$\mathrm{Accumulated}\;\mathrm{release}\;\mathrm{rate}\;(\%)\;=\;[C_{i}\times V+\sum_{i=1}^{n}(C_{i-1}\times V_{i-1})]/10\times 100\%$$
where C_i_ is the ith release pool FITC-INS concentration, i is the number of sampling times, V is the release pool capacity, and V_i_ is the volume required at each sampling. n is the total number of sampling times and C_0_ is 0 IU.

#### 2.12.2. Insulin Release in Solutions of Different Glucose Concentrations

The glucose-responsive SF MNs were pricked into fresh rabbit skin, fixed, and placed in a 12 mL transdermal drug release cell, and the designed release solutions were four concentrations of 0, 100, 200 and 400 mg/dL of glucose solution. Hence, 1 mL of samples was taken at 1, 2, 4, 8, 12, 24, and 36 h for testing, and a fresh 1 mL of release solution was added after each sample to ensure that the total volume in the release cell was constant at 12 mL. The formula for calculating the cumulative release rate is given in Equation (3).

### 2.13. In Vivo Drug Release Properties

#### 2.13.1. Changes in BG Content in Rats after MNs

(1) Diabetic rat modeling

Diabetic rats were induced with Streptozotocin (STZ), and the BG concentration was tested by a BG meter. The modeling was successful if the BG was stable and higher than 300 mg/dL [41].

(2) Experimental grouping

Group I: experimental diabetic rats group with MNs: diabetic rats 1, 2 and 3 were anesthetized with an appropriate amount of ether, shaved, attached with glucose-responsive SF MNs (human insulin content 10 IU/tablet), and bandaged.

Group II: control diabetic rats group injected: diabetic rats 4, 5 and 6 were directly injected with human insulin (10 IU/each).

Group Ⅲ: experimental regular rats group with MNs: rats 7, 8, 9 with glucose-responsive SF MNs (10 IU/tablet).

Group Ⅳ: control regular rats group with blank MNs: rats 10, 11, 12 affixed with pure SF MNs (0 IU/tablet).

(3) Sampling and testing (BG and insulin content)

Blood was collected intravenously from 12 rats separately, and the BG concentration was tested by a BG meter. 20 μL of blood was taken, diluted 10 times by PBS solution, centrifuged at 3000 rpm for 10 min, and the serum was separated. The diabetic rats 1–3 and normal rats 7–9 were attached with glucose-responsive SF MNs, and normal rats 10–12 were attached with pure SF MNs to start fasting; diabetic rats 4–6 were directly injected with insulin (10 IU/each) and fasting was started. Blood samples were taken at 1, 2, 3, 4, and 6 h to measure BG concentration and insulin concentration. After 6 h, feeding was started, rats were monitored, and blood samples were taken at 1, 2, 4, and 6 h after feeding. Blood samples were taken at 1, 2, 4, and 6 h after feeding to measure BG and insulin concentrations.

#### 2.13.2. Changes in Insulin Content in Rats after MNs

Insulin levels in rats were determined using a chemiluminescence assay kit. The samples to be measured were equilibrated at room temperature for 30 min. The luminescent substrates A and B were mixed equally and prepared for use. The pieces were added to the 96-well assay plate. Then, 50 μL of standards of different concentrations were added to the first five wells, 50 μL of samples were added to the remaining wells, and 50 μL of enzyme conjugate was added to each well. The wells were shaken in a shaker to mix the liquid in the wells and placed at 37 °C for 60 min. The wells were washed 5 times with washing solution using a plate washer. 100 μL of the mixed luminescent substrate was added and reacted in a luminometer (LUMO, Zhengzhou Anto Bioengineering Co., Ltd., Zhengzhou, China) for 5 min. The luminescence intensity was detected, the standard curve of the luminescence method was plotted, and the insulin content in rats was calculated.

The relative bioavailability of insulin was calculated by the insulin curve integration method with the following formula.
RBA(%) = (AUC_MN_ × Dose_IP_)/(AUC_IP_ × Dose_MN_) × 100%(4)
where AUC_MN_ indicates the area under the blood concentration-time curve after insulin-laden MNs administration, AUC_IP_ indicates the area under the blood concentration-time curve after insulin injection, and Dose_IP_ and Dose_MN_ both suggest the dose administered.

## 3. Results

### 3.1. Characterization of Cationized SF Properties

The Zeta potential of the polylysine-modified SF in Figure 2a gradually changed from the negative value of pure SF (around - 7 mv) to the positive value of grafted SF as the polylysine content increased from 0% to 15%. Further, expanding the range to 20% polylysine, the gelling phenomenon of the SF solution was too severe to continue the experiment. Therefore, a 10% polylysine graft-modified SF solution was used in the modification experiments, and the zeta potential value changed to about + 6.3 mv after its modification.

As shown in Figure 2b, the IR absorption spectrum of the 10% polylysine graft-modified SF showed a more substantial absorption peak at 3439 cm^−1^, the characteristic peak belonging to the amino group (3500–3300 cm^−1^), indicating the successful introduction of more amino groups. Furthermore, the comparison revealed that the wave number of the characteristic peak of the amide bond of the SF changed. A small peak appeared at the typical peak position at 1635 cm^−1^. This indicates the formation of some β-fold structure in the wave number range of amide I. Comparing the IR spectra of the SF before and after cationization, it can be shown that the polylysine grafting modified SF did increase the amino group, and the polylysine was successfully grafted to the SF.

### 3.2. Characterization of MNs

As shown in Figure 3a, the MNs array in i–ii is structured with the same length and uniform arrangement, and the size of the MNs is approximately 430 μm. The MNs in iii–iv are not damaged, the tip is intact, and normal penetration can be ensured.

### 3.3. Mechanical Properties

In order to compare the differences in mechanical properties between glucose-responsive MNs and pure SF MNs, the effect of Gox incorporation on the mechanical properties of MNs was investigated. Figure 3b shows that the pure SF MNs had the highest fracture strength with high mechanical properties. In contrast, in the glucose-responsive MNs, the force was decreased because of the increased flexibility caused by the grafting of polylysine. The addition of Gox leads to another decrease in MNs fracture strength. However, the MNs strength was still more than 1.25 N, which was sufficient to pierce the skin due to the relatively sharp MNs head. In order to guarantee the MNs have a high sensitivity to BG concentration even with a significant fracture strength, the addition of 4 mg/mL of Gox to the MNs was chosen in this paper.

### 3.4. Conformation and Aggregation Structures

The microstructural changes of glucose-responsive SF MNs and pure SF MNs were observed. Figure 4a demonstrates that glucose-responsive SF MNs have distinct and sharp absorption peaks at 1635 cm^−1^ (amide I), 1540 cm^−1^ (amide II), and 1235 cm^−1^ (amide III), while pure SF MNs have distinct and sharp absorption peaks. These are the characteristic peaks of the β-fold structure. As can be seen from Figure 4b, the pure SF MNs showed an extremely prominent absorption peak at 20.7° and a diffraction peak at 12.2°, pointing to the formation of a partially crystalline structure of the SF protein in the MNs. In contrast, the glucose-responsive SF MNs only had a more significant bun peak, indicating weak crystallinity. This may result from covalent cross-linking to form a hydrogel that inhibits the crystallization of the filamentous proteins. These non-crystalline filamentous proteins are expected to have a relatively large swelling, allowing for the release of the drug.

### 3.5. Responsive Performance

Figure 5a shows the dissolution of MNs s at different pH values. After reaching the swelling equilibrium in different pH buffered (pH = 4, 7, 9) solutions for one day, the degree of swelling of pure SF hydrogels showed an increasing tendency with the increase of pH. The MNs immersed in pH = 9 solution had the greatest swelling extent, exhibiting a sensitive swelling property to alkaline pH, resulting from the negative charge of the SF. The higher the pH value, the more negative the amount. The repulsion of negative charges increases the swelling of MNs when they are alkaline.

Meanwhile, glucose-responsive SF MNs showed acidic pH sensitivity. In contrast to pure SF MNs, the swelling properties are enhanced in an acidic environment. The reverse situation is exhibited in alkaline environments. The reason for this is that grafted SF is positively charged. The pH is smaller, the more positive charge there is. The repulsion of the positive charges increases the solubility of the MNs when it is acidic. As seen in Figure 5b, the responsiveness of the gel interior to different glucose levels was evident. As the glucose concentration increased from 0 mg/dL to 400 mg/dL, the pH within the gel decreased from 7 to 4.6. As time passed, the pH gradually reduced until it reached equilibrium. At the same time, the external solution’s pH remained unchanged, demonstrating that the overall pH of the solution did not change with the internal pH. It was also not revealed whether hydrogen ions diffuse to the outside of the gel due to the high pH of the whole solution, affecting the internal pH and thus reducing the pH response performance. This sets the stage for the subsequent development of glucose-responsive SF MNs. The concentration level signal of glucose inside the MNs translates into pH stimulation. In contrast, the external pH of the MNs remains unchanged, causing no additional acidic stimulation to the skin.

### 3.6. In Vitro Drug Release Properties

We prepared a fluorescent value-concentration standard curve of fluorescently labeled human insulin to calculate the released drug concentration, shown in Figure 6a. 

#### 3.6.1. In Vitro Drug Release Properties in Buffered Solutions of Different pH Values

As shown in Figure 6b, the glucose-responsive SF MNs showed different rates of insulin release in three different pH (pH = 4, 7, 9) buffer solutions. The rate of insulin release was fastest at pH = 4, which was significantly quicker than the rate of insulin release at the other two pH conditions. Pure SF MNs were used as a control group without drug loading, which did not have drug release in its release profile, excluding the possible changes in fluorescence values caused by SF itself. The isoelectric point of insulin is 5.4. Insulin has a positive charge at pH = 4. Glucose-responsive SF MNs have positive control of their own, generate co-charge repulsion, and increase the rate of insulin release. Insulin is negatively charged at pH = 7, there is no charge repulsion, so insulin drug release can only rely on the dissolution of the MNs. The slow dissolution of the MNs releases the insulin drug slowly. The lowest MNs swelling and lowest drug release accumulation rate were observed at pH = 9. After the MNs were dismissed for 12 h, the drug release of MNs insulin in a pH = 4 environment reached 44.1%, while the drug release in a pH = 9 climate was only 26.4%. The release of MNs gradually decreased in the later period and finally reached the drug release equilibrium.

#### 3.6.2. In Vitro Drug Release Properties in Buffered Solutions of Different pH Values

Figure 6c and reveals that glucose-responsive SF MNs have different rates of insulin release in solutions with varying glucose levels. The rate of insulin release was fastest in the high glucose concentration solution (400 mg/dL). Otherwise, the rate of insulin release was slowest in the no-glucose concentration solution (0 mg/dL). As seen in Figure 7d,e, the MNs tip is almost dissolved after administration. The dissolving structure allows MNs to absorb body fluids and expand, creating a dynamic cycle to facilitate insulin release from the base. Figure 6f displays the variation of MNs pore size in different glucose concentration solutions. The MNs section showed a multivacancy cross-linked mesh structure in cross-section, exhibiting a large swelling degree. As the glucose concentration in the solution increases, the pore size of the MNs increases, indicating that the MNs have glucose-responsive properties, capable of transforming the glucose concentration signal into a pH signal and stimulating a micro swelling response to glucose.

### 3.7. In Vivo Drug Release Properties

#### 3.7.1. Changes in Blood Glucose Content in Rats after MNs

The MNs attachment method is provided in Figure 7a. The results in Figure 7b reveal blood glucose changes in the experimental and control groups 6 h before and after meals. Normal rats maintained normal BG levels at 7.4 mmol/L ± 1.3 mmol/L during pre and postprandial meals. The BG level of SD diabetic rats in the MNs patch group decreased slowly from 22.4 mmol/L to 11.7 mmol/L before the dinner, and the BG gradually normalized. The BG level in the injection group decreased rapidly from 22.4 mmol/L to 7.8 mmol/L within 1 h and continued to decline subsequently. After the meal, the BG of SD diabetic rats in the postprandial MNs patch group rose to 21.7 mmol/L. After 2 h, there was a decreasing trend, and the BG level was 15.3 mmol/L after 6 h, and gradually returned to average BG concentration; in the injected group, the BG increased rapidly to 33.4 mmol/L and showed a slow decline at 1 h. After 6 h, the BG concentration was still as high as 26.9 mmol/L.

#### 3.7.2. Changes in Human Recombinant Insulin Content in Rats after MNs Application

Figure 7c indicated that direct injection of human insulin in diabetic rats led to a dramatic rise in human recombinant insulin in the body and was significantly higher than the human recombinant insulin content in rats with MNs. The level of human insulin peaked after 1 h and declined continuously afterward. However, a slow rise in human insulin levels in diabetic rats, after patching, peaked around 4 h before a meal. After meals, human insulin levels presented a trend of decreasing, then increasing, and finally dropping. The relative utilization of human recast insulin in MNs was calculated to be 75%, based on the human recombinant insulin curve integration method, showing a trend of higher utilization efficiency. It is possible to ensure the effective release and utilization of human insulin from MNs. In normal rats, after using a patch, human insulin was slightly increased in vivo, followed by a decline. This indicates that glucose-responsive SF MNs release less insulin at average BG concentrations. An increase in insulin release was also demonstrated in response to a rise in postprandial glucose concentration, signifying that glucose-responsive SF MNs achieved responsive freedom in the normal rat state.

## 4. Discussion

The earlier insulin MNs that were studied had soluble tips. After the patient applies the MNs, the immediate release of the drug from the front can satisfy the need for blood sugar reduction. However, SF also dissolves into the body. SF has good biocompatibility, but the effects caused by long-term accumulation in the body are unpredictable. Dissolving MNs can avoid this risk but they do not control the release of the drug, while excessive discharge may have the risk of causing hypoglycemia. Innovative, responsive MNs bridge this gap. In this study, cationized SF was obtained by graft copolymerizing SF with polylysine as MNs substrate, which gave the MNs acidic pH sensitivity.

The blood level of insulin will significantly affect the BG concentration. In order to prevent hypoglycemic shock caused by an overdose of insulin, the MNs system is required to control the rate of insulin release autonomously and be able to respond to BG concentration. Glucose-responsive systems using Gox are always associated with pH-sensitive materials. It is possible to introduce Gox, which converts glucose to gluconic acid, creating an enzyme-mediated change in the pH environment. As illustrated in Equation (5) below, the regulation of the change in glucose concentration leads to the production of gluconic acid, which provides an acidic environment.
Β-D-glucose + O_2_ → D-Gluconic acid + H_2_O_2_(5)

This opens up the possibility of developing glucose-responsive SF MNs. It can be changed from pH-sensitive to glucose-sensitive, which enables controlled release of the drug at different BG concentrations. Ye et al. [42] reported MNs made of polymeric nanovesicles composed of Gox, α-amylase, and glucoamylase with hyaluronic acid. Ullah et al. [16] made MNs glucose responsive by adding NaHCO_3_ and Gox, where NaHCO_3_ acts as a pH-sensitive component, to enable controlled insulin release. Furthermore, previous studies have demonstrated that silk protein can be used as an enzyme immobilization substrate. Lu et al. [43] revealed that the activity of Gox was hardly reduced when the Gox-loaded SF membrane was stored at 37 °C for 10 months. This ensures the Gox activity in the modified SF MNs and provides a theoretical basis for the response stability of the MNs.

Glucose-responsive SF MNs were obtained by adding HRP, H_2_O_2_, Gox, and insulin to the polylysine-modified SF solution. The MNs swell but do not dissolve when inserted into the skin. MNs have high mechanical strength, which can pierce the human epidermis under dry conditions and reach the dermis, creating a minimally invasive array of microchannels on the skin surface and enabling easy transdermal drug delivery. MNs piercing the skin rehydrate by absorbing interstitial fluid to form a semi-solid hydrogel with a porous cross-linked mesh structure and release internal insulin into the skin tissue in a controlled manner through the porous 3D mesh structure.

The principle of transdermal drug release from glucose-responsive SF MNs under different pH and glucose concentration conditions was investigated by in vitro experiments. As Figure 8 shows, the transdermal delivery capacity of insulin decreases as the pH increases. This is because polylysine contains a large number of amino groups. After grafting successfully, the modified serine protein molecule has a positive charge. The solution includes a large amount of H^+^ at low pH. Hydrogen ions enter the hydrogel to bind to the polylysine amino group, which positively charges the modified serine protein. Same-charged phase repulsion occurs, and it dissolves the MNs. Since MNs contain a large number of -OH, -COOH, -NH_2_, and other groups, the weakly ionized group -COOH in the SF fraction undergoes proton transfer at different pH. In acidic media, -OH and -NH_2_ groups are easily protonated to -H_2_O^+^ and -NH_3_^+^, which causes ionic and intermolecular interactions with water molecules and enhance the hydrophilic properties of the macromolecular chains inside the MNs. The lower the pH, the more H^+^, and with the greater the repulsion of the same charge, swelling degree increased. The number of protonated -OH and -NH_2_ groups decreased gradually with the increase in pH. The hydrophobicity of MNs is strengthened, the concentration of H^+^ in solution gradually decreases, and the repulsion of the same charge becomes smaller with less and less dissolution. As the glucose concentration increases, the transdermal drug diffusion capacity increases. The reason for this is the oxidation of glucose to gluconic acid under the action of Gox, which changes the local pH environment of the MNs and gradually increases the H^+^ concentration in the solution, promoting MNs lysis. Hence, transdermal insulin release is improved.

An SD diabetic rat model was further established while using normal rats as a control group, testing the glucose responsiveness of MNs in vivo. A gradual decrease in BG level in diabetic rats after MNs application, dropped to 11.7 mmol/L after 6 h, indicates the MNs can effectively release insulin and lower blood sugar. The BG level in normal rats after MNs only slightly decreased and maintained stable. The comparison demonstrates that MNs have fast insulin release rate and large release volume at high BG concentrations with insulin-controlled release capability. Unlike direct insulin injection, modified SF MNs cause no hypoglycemia problems with only a slight insulin release at low BG concentrations. This indirectly proves the advantages of MNs for the sustained release of drugs. After rats were fed, it was noted that all three groups had elevated BG at 1 h after the meal, with the most significant increase in the experimental group of rats with insulin injection.

For this reason, the digestion of food intake after eating can cause a sudden rise in BG. The insulin in the injection group has been degraded and challenging, making it difficult to lower blood glucose. Two hours after the meal, the BG of diabetic rats started to decrease gradually after the MNs, resulting from insulin release within the MNs. The comparison revealed that two normal rats’ blood glucose remained stable after the MNs, which reinforced the idea that these developed glucose-responsive SF MNs are indeed glucose-responsive with different insulin release rates at different BG concentrations.

In summary, we prepared SF MNs with pH and insulin responsiveness for biomedical applications. Cationized SF was obtained by graft copolymerization modification with polyllysine. Crosslinking with HRP/H_2_O_2_ formed a pH-responsive cationic gel. The use of super-swelling polymers as a foundation material to SF MNs ensures a high swelling rate [44]. The internal drug release can be controlled by adjusting the pore size [45,46]. Gox [47,48] and phenylboronate polymers [49] are widely used as glucose-sensitive components to prepare glucose-dependent insulin release systems [50]. We prepared glucose-responsive MNs by adding insulin and Gox to the gel. The MNs pierce the skin and absorb interstitial fluid to form a semi-solid hydrogel, the pore size of which in the porous 3D mesh structure increases with elevated glucose concentration, facilitating the controlled release of insulin by MNs. Compared to conventional injections, in vivo studies have indicated that glucose-responsive SF MNs can achieve intelligent control of BG levels before and after feeding in diabetic SD rats.

## 5. Conclusions

In this paper, cationized SF was obtained by graft copolymerization modification of polylysine, and glucose-responsive SF MNs were prepared by adding insulin and Gox to this base material. The breaking strength of a single MN is more than 1.25 N, which can penetrate the human epidermis smoothly under dry conditions and achieve transdermal drug delivery. The MNs pierce the skin and absorb interstitial fluid to form a semi-solid hydrogel, which controls the release of internal insulin into the skin tissue through a porous 3D mesh structure. In vitro studies indicated that MNs exhibit excellent pH-responsive solubility and glucose-responsive insulin recovery. In vivo studies have shown that MNs control BG levels before and after feeding in diabetic SD rats as compared to conventional injections. This study demonstrates the potential application of glucose-responsive SF MNs in treating diabetes mellitus through transdermal insulin uptake. The research results of this paper lay a certain foundation for the intelligent transdermal release of insulin.

## Figures and Tables

**Figure 1 biomimetics-08-00050-f001:**
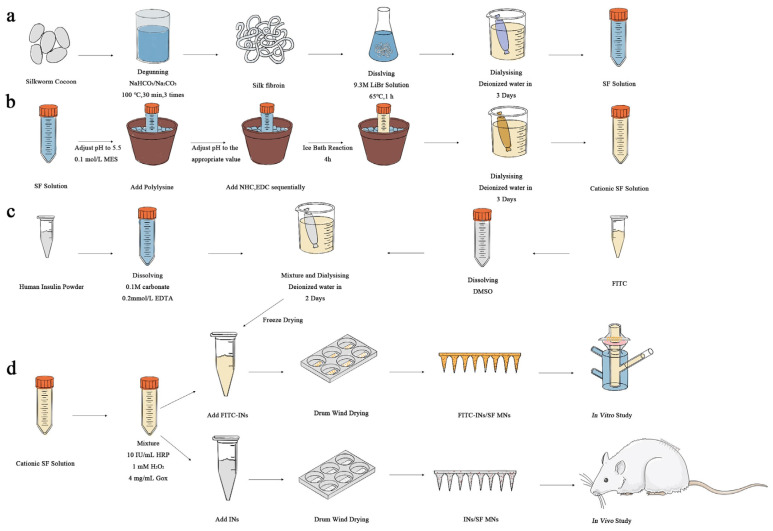
Preparation of glucose-responsive SF MNs (**a**) preparation of SF; (**b**) cationization of SF; (**c**) preparation of FITC-INs; (**d**) preparation of glucose-responsive SF MNs with in vitro/in vivo drug release studies.

**Figure 2 biomimetics-08-00050-f002:**
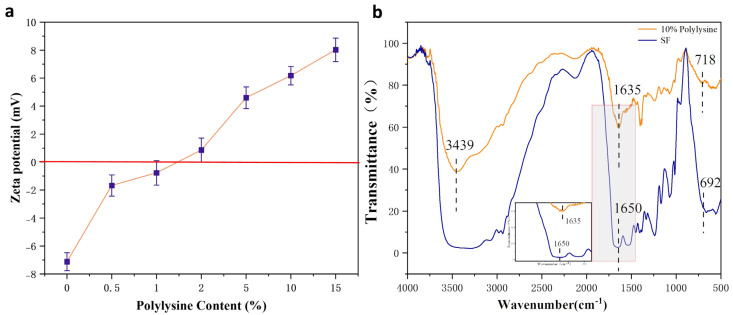
Results of polylysine grafted SF (**a**) zeta potential; (**b**) FTIR.

**Figure 3 biomimetics-08-00050-f003:**
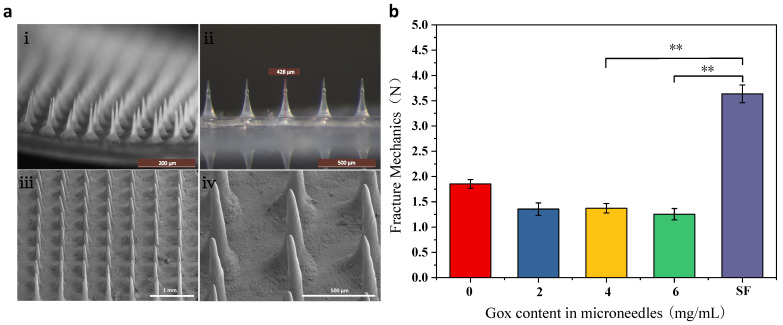
Morphology and mechanical properties of MNs (**a**) morphology of glucose-responsive SF MNs (**i**,**ii**): body microscope photographs; (**iii**,**iv**): scanning electron microscope photographs; (**b**) mechanical properties of glucose-responsive SF MNs and pure SF MNs. **: *p* < 0.01.

**Figure 4 biomimetics-08-00050-f004:**
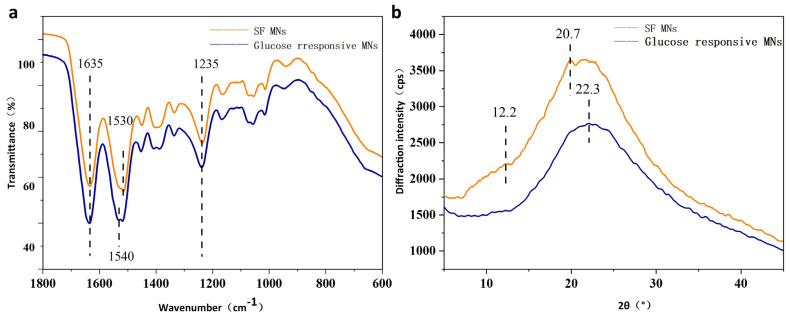
Structures of glucose-responsive SF MNs and pure SF MNs (**a**) FTIR; (**b**) X-ray diffraction curves.

**Figure 5 biomimetics-08-00050-f005:**
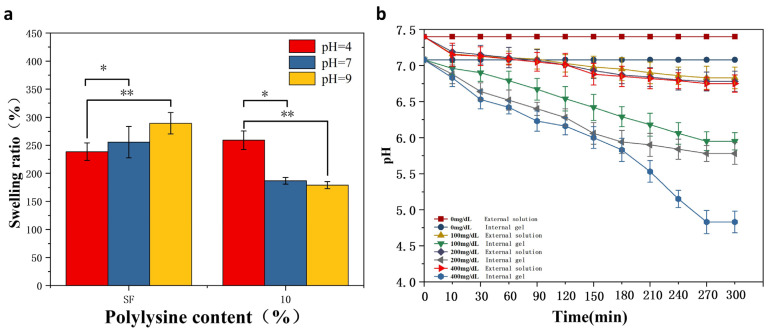
pH/glucose responsiveness of MNs (**a**) dissolution of glucose-responsive SF MNs and pure SF MNs in buffered solutions of different pH. (**b**) Changes in internal and external pH of gels at different glucose concentrations. *: *p* < 0.05; **: *p* < 0.01.

**Figure 6 biomimetics-08-00050-f006:**
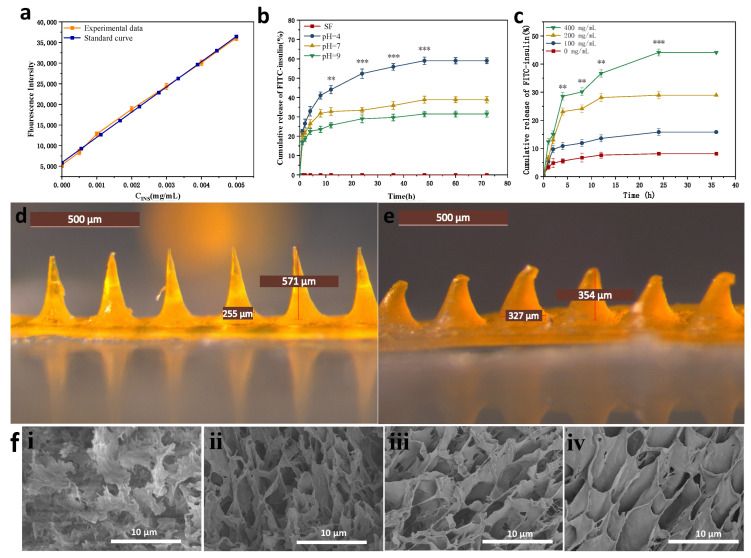
In vitro drug release curves and morphology of MNs before and after drug release (**a**) fluorescence value-concentration standard curve of fluorescently labeled insulin (**b**) cumulative insulin release rate curves of MNs in different pH buffers; (**c**) cumulative insulin release rate curves of MNs in different glucose concentration solutions (**d**) morphology of MNs after completion of MNs fabrication; (**e**) morphology of MNs after 36 h drug release by piercing into rabbit skin; (**f**) Cross-sectional electron micrographs of MNs in different glucose concentration solutions (**i**): 0 mg/dL; (**ii**): 100 mg/dL; (**iii**): 200 mg/dL; (**iv**): 400 mg/dL. **: *p* < 0.01; ***: *p* < 0.001.

**Figure 7 biomimetics-08-00050-f007:**
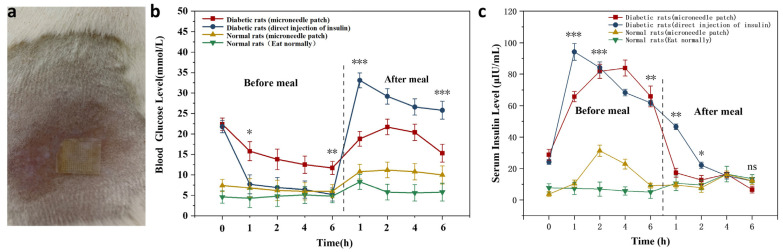
Experimental results in rats (**a**) MNs patched on the mouse’s back. (**b**) The change curve of BG in rats before and after meals following MNs (**c**) The change curve of insulin in rats before and after meals following MNs. *: *p* < 0.05; **: *p* < 0.01; ***: *p* < 0.001.

**Figure 8 biomimetics-08-00050-f008:**
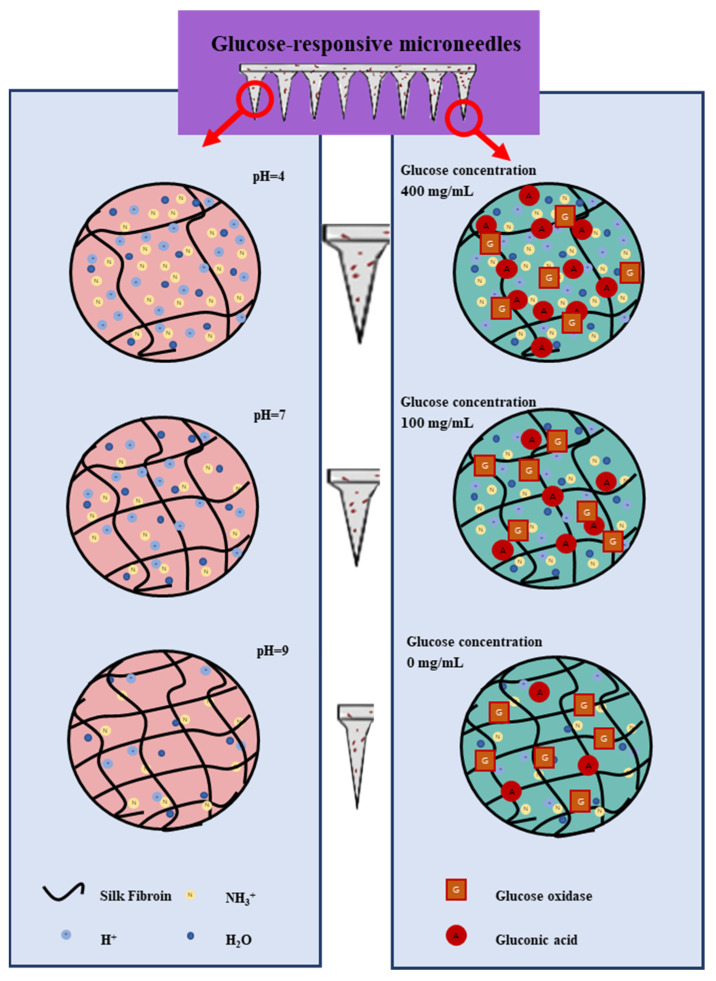
pH-responsive and glucose-responsive drug release mechanism of MNs.

## Data Availability

Not applicable.

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
