# Peer review of "Glucose-Responsive Silk Fibroin Microneedles for Transdermal Delivery of Insulin"

_biomimetics, 2023, doi:10.3390/biomimetics8010050_

Round 1

Reviewer 1 Report

Regarding the manuscript (biomimetics-2126909) entitled:

“Glucose-responsive Silk Fibroin Microneedles for Transdermal Delivery of Insulin”

Comments to the Author

General comment

The manuscript described MNs development from polylysine-modified cationized silk fibroin (SF) for responsive transdermal insulin delivery. The manuscript, in general, is well written and should be published after considering the following comment:

1.       Abstract needs more numerical information

2.       Figure 1: please add arrows to show scheme directions.

2.5.  Preparation of Glucose-responsive SF MNs: No details about MN mold

3.        Figure 3 scale bars of ai and aii are not clear

4.        Figure 5 the legend of internal gel is missing the letter i

5.        Figure 6a please add an error bar for each point

6.      Figure 7: an image for Microneedle penetration is required to confirm that the MN mechanical properties can penetrate skin layers similar to this article DOI: 10.1007/s13346-015-0237-z

7.       Discussion section should be improved with a comparison to previous studies

Author Response

We appreciate the reviewer’s positive evaluation of our work. Thanks very much for taking your time to review this manuscript. We really appreciate all your generous comments and suggestions! We have responded to each of the comments made by the reviewer one by one. Please see the attachment.

Reviewer 2 Report

Reviewer: Presents a comprehensive work on the development of the manuscript " Glucose-responsive Silk Fibroin Microneedles for Transdermal Delivery of Insulin". The analysis done is detailed and well-organized, which gives plenty of its merits. However, the discussion is a bit poor, the authors should elaborate on the discussion of the results obtained. Additionally, the main problem with this article lies in the experimental design and statistical analysis.

The appropriate reasons for reducing the experimental samples are not mentioned in detail.

The assumptions for ANOVA should be verified before running the ANOVA test. The assumptions of normality and normal variance would be very difficult to test with small sample sizes

Common comments are as follows:

1. The English needs to be improved and please double-check grammar and spelling.

2. The introduction section should be properly revised.

3. How many measurements were performed to determine the pore size of scaffolds?

4. Significant (p-values) should be prepared according to the statistical analysis.

5. I do not find any significant data.  Please give additional quantitative data with statistical value. as depicted in Figure 6,7, significant. This figure does not reflect in statistics, correct them.

6. Conclusions may be more appealing if authors include key insights and future prospects, rather than just summarizing the results.

Author Response

(The authors gave the same response as above.)

Round 2

Reviewer 1 Report

no comments